# Phenotypic and Genotypic Characterization of ESBL and *Amp*C β-Lactamase-Producing *E. coli* Isolates from Poultry in Northwestern Romania

**DOI:** 10.3390/antibiotics14060578

**Published:** 2025-06-05

**Authors:** Anca Rus, Iulia-Maria Bucur, Kalman Imre, Andreea Talida Tirziu, Andrei Alexandru Ivan, Radu Valentin Gros, Alex Cristian Moza, Sebastian Alexandru Popa, Alexandra Ban-Cucerzan, Emil Tirziu

**Affiliations:** 1Faculty of Veterinary Medicine, University of Life Sciences “King Mihai I” from Timisoara, Calea Aradului 119, 300645 Timisoara, Romania; mekker.anca-sm@ansvsa.ro (A.R.); kalman_imre27@yahoo.com (K.I.); andreialexandru.ivan.fmv@usvt.ro (A.A.I.); valentingros@usvt.ro (R.V.G.); sebastian.popa@usvt.ro (S.A.P.); alexandracucerzan@usvt.ro (A.B.-C.); emiltarziu@yahoo.com (E.T.); 2Faculty of Medicine, “Victor Babes” University of Medicine and Pharmacy, Piata Eftimie Murgu 2, 300041 Timisoara, Romania; andreea.tirziu@umft.ro

**Keywords:** extended-spectrum beta-lactamase (ESBL), *Amp*C, multidrug resistance (MDR), *E. coli*, antibiotic resistance genes, penetrance

## Abstract

Background/Objectives: The widespread use of antibiotics in animal husbandry has led to an increase in antimicrobial-resistant *Escherichia coli*, particularly strains producing extended-spectrum β-lactamases (ESBL) and *Amp*C β-lactamases. This study aimed to isolate and characterize such strains from fecal samples of broiler chickens (n = 71) and slaughtered turkeys (n = 31) in northwestern Romania. Methods: Antimicrobial susceptibility testing and PCR were used to evaluate phenotypic resistance patterns and detect the presence of resistance genes (*Amp*C, *bla*Z, and *bla*TEM). Results: The results showed that 55% of turkey and 61% of broiler isolates were presumptive ESBL/*Amp*C producers. Among all isolates, 50% were classified as extensively drug-resistant (XDR), 44% as multidrug-resistant (MDR), and only 6% were fully susceptible. Gene detection revealed an overall prevalence of 44.2% for *Amp*C, 72.7% for *bla*Z, and 58.1% for *bla*TEM, yielding a total penetrance of 51.09%. The diagnostic odds ratio (DOR) values, ranging from 0.67 to 81, suggest the efficacy of the antibiotic susceptibility testing method used in detecting the presence of these resistance genes. Conclusion: Overall, these findings highlight a significant burden of antimicrobial-resistant, poultry-associated *E. coli* strains, warranting stricter antimicrobial stewardship.

## 1. Introduction

Bacterial resistance to antimicrobial agents is a major cause of increased mortality and morbidity worldwide. The remarkable adaptability exhibited by bacterial strains is the result of multiple interactions between antimicrobial substances and these strains. In this context, the widespread use of antibiotics accelerates the evolution of bacterial strains, leading to antibiotic resistance [1,2,3,4].

The most common mechanism of antimicrobial resistance (AMR) in *E. coli* is the inactivation of β-lactams antibiotics through the production of enzymes specifically extended-spectrum β-lactamases (ESBLs) and transferable *Amp*C β-lactamases. These enzymes can inactivate not only penicillins, but also first, second, or third-generation cephalosporins [5,6].

The class of β-lactam antibiotics, which include penicillins, cephalosporins, and carbapenems, act as inhibitors of bacterial cell wall synthesis and represents the most widely used class for treating bacterial infections, due to their broad spectrum of activity against both Gram-negative, as well as for Gram-positive pathogens [5,6,7,8]. These antibiotics share the presence of a four-carbon atom ring in their molecular structure, known as the β-lactam ring. Through hydrolysis, β-lactamases break open this ring, deactivating the antibacterial properties of the molecule. Clavulanic acid is a β-lactamase inhibitor commonly used in combination with β-lactam antibiotics (e.g., amoxicillin-clavulanate) to restore their efficacy by binding irreversibly to β-lactamases. Cephalosporins are divided into generations based on their antimicrobial spectrum, with later generations (e.g., third- and fourth-generation cephalosporins like cefotaxime and cefepime) exhibiting enhanced activity against Gram-negative organisms. Carbapenems (e.g., imipenem, meropenem) are considered last-resort antibiotics due to their broad-spectrum activity and resistance to most β-lactamases, although resistance through carbapenemase production has emerged. Understanding these classes and their mechanisms of resistance is crucial in evaluating the antimicrobial profiles of bacterial isolates [9,10]. Penicillins and cephalosporins are included in many treatments; therefore, characterizing the genes and enzymes that confer antibiotic resistance to these classes of substances in *E. coli* strains plays an important role in selecting appropriate therapies [11,12].

The emergence and spread of antimicrobial resistance (AMR) in *Escherichia coli* from poultry sources has become a growing concern due to the intensive use of antibiotics in animal production, often as prophylactics or growth promoters. Several studies have documented the high prevalence of ESBL and *Amp*C-producing *E. coli* in poultry across Europe and other regions, with third-generation cephalosporins, such as cefotaxime and ceftazidime, being critically important antimicrobials in both veterinary and human medicine [13,14,15]. The monitoring of AMR in poultry is essential due to the zoonotic potential of resistant *E. coli* strains, which can be transmitted to humans via the food chain or the environment [16,17].

ESBL and *Amp*C β-lactamases are two major mechanisms by which *E. coli* develop resistance to β-lactam antibiotics, especially cephalosporins. While ESBLs, such as those encoded by *bla*TEM, hydrolyze a wide range of β-lactams and are often plasmid-borne, *Amp*C β-lactamases—encoded by the *Amp*C gene—can be chromosomal or plasmid-mediated, with increasing detection in food-producing animals. The inclusion of *bla*Z, though traditionally associated with *Staphylococcus aureus*, in this study also reflects recent findings of horizontal gene transfer and its emerging presence in Gram-negative organisms [18].

The selection of *Amp*C, *bla*TEM, and *bla*Z genes in this study was based on both their clinical relevance and their reported prevalence in poultry from different geographical regions, including neighboring countries such as Hungary, Poland, and Italy, where similar resistance patterns have been observed [19,20,21]. Furthermore, limited data were available from northwestern Romania, making it necessary to fill this surveillance gap and assess the local resistance landscape [22].

Detecting these genes provides a dual benefit: it allows us to monitor resistance trends and evaluate the effectiveness of phenotypic susceptibility testing by comparing it with genotypic data. This is particularly important in One Health surveillance frameworks, where early identification of resistance genes can help guide targeted interventions in both veterinary and human health sectors [23,24].

Characterizing ESBL/*Amp*C-producing *E. coli* in poultry, therefore, contributes not only to national AMR surveillance but also to global efforts to control the spread of resistant pathogens. As such, this study seeks to provide a detailed phenotypic and genotypic characterization of resistant *E. coli* strains isolated from poultry in northwestern Romania, with a focus on genes that are both prevalent in similar contexts and carry significant clinical implications.

Plasmid-mediated ESBL genes are of great importance in public health and can easily be transferred between bacterial species. Some ESBL genes are mutant derivatives of specific, plasmid-mediated β-lactamases (e.g., *bla*TEM), while others are mobilized from bacteria present in the environment [22,25,26].

The production of *Amp*C β-lactamases can be due to both chromosomal *Amp*C genes and the presence of plasmid-mediated genes in *E. coli* strains. However, strains producing *Amp*C enzymes may undergo chromosomal mutations that result in the overexpression of chromosomal genes and the secretion of these enzymes in large quantities [26,27].

The prevalence of *E. coli* ESBL/*Amp*C strains isolated from broiler chickens and chicken meat showed a slight decrease in the EU member states. In 2022, the prevalence was 34.9% in broiler chickens and 29.4% in chicken meat, a slight decrease compared to the data reported in 2020: 38% and 30.6%, respectively. Additionally, the prevalence of *E. coli* ESBL/*Amp*C strains in fattening turkeys was 32.1%, which is much lower than the 47% recorded in 2016. The trend of decreasing antibiotic-resistant *E. coli* strains isolated from turkeys was observed in six countries: Austria, France, Hungary, Poland, Spain, and Sweden. This decrease was largely attributed to the reduction in the irrational use of antibiotics and the ban on the off-label use of antibiotics in broiler chickens and fattening turkeys [28].

The latest European report on antimicrobial resistance of zoonotic strains, from animals and humans, for the years 2021–2022, shows that for *E. coli* strains, the occurrence of isolates with the ESBL/*Amp*C phenotype ranged from 24.6% (Latvia) to 97.7% (Germany) in broiler chickens [28]. Regarding fattening turkeys, the occurrence of these phenotypes ranged from 47.1% (Romania) to 97.5% (Spain). The overall prevalence of ESBL/*Amp*C-producing strains was 32.1% in fattening turkeys and 34.9% in broiler chickens.

According to the EU Protocol, Version 7/2019, for isolating *E. coli* strains that produce ESBL and *Amp*C β-lactamases from a fecal sample, only one strain is isolated and characterized [23,29,30,31]. If both the ESBL and *Amp*C phenotypes are found in the same fecal sample, the likelihood of detecting one phenotype or the other depends on the abundance of the respective phenotype in the analyzed sample. However, in all EU countries’ reports, the occurrence of *E. coli* strains producing ESBL exceeds that of *Amp*C-producing strains [28].

In this context, the study aimed to isolate *E. coli* strains producing ESBL/*Amp*C from fecal samples collected from broiler chickens and slaughtered turkeys between 2020 and 2022, to characterize the phenotypic antimicrobial resistance of the isolates, to study the phenomenon of “multidrug resistance” (MDR) in these strains and, also, to determine the presence of antimicrobial resistance genes (*Amp*C, *bla*Z and *bla*TEM). The main goal was to determine the presence of these genes in strains that exhibited AMR, as well as to assess the correlation between the phenotypic and genotypic aspects of antimicrobial resistance in the isolated *E. coli* strains.

## 2. Results

Out of the 102 fecal samples, 31 were collected from turkeys with a slaughter age between 13 and 21 weeks, while 71 samples were collected from broiler chickens aged between 36 and 50 days. On MacConkey agar, with or without cefotaxime, the isolated *E. coli* strains produced flat, dry, pink, and non-mucoid colonies, accompanied by a dark pink halo resulting from the precipitation of bile salts. The strains were morphologically analyzed and biochemically tested, confirming the species of *E. coli*. As presented in Table 1, from the total of 102 isolates, 60 grew on both media (MacConkey agar and MacConkey agar supplemented with cefotaxime), indicating that they were likely ESBL producers. On the other hand, the rest of the isolates (n = 42) grew only on MacConkey agar, suggesting they were commensal *E. coli* strains with lower antibiotic resistance, since their growth was inhibited by cefotaxime in the medium.

The proportion of presumptive ESBL/*Amp*C-producing strains was 55% out of the total strains isolated in turkeys and 61% out of the total isolates in broiler chickens (*p* > 0.05) (Table 1).

### 2.1. Antimicrobial Susceptibility Testing (AST) Results

The results showed that 25 (81%) of the strains isolated from turkeys were resistant to ampicillin, 24 (77%) strains to tetracycline, 23 (74%) strains to chloramphenicol and sulfamethoxazole, 22 (71%) strains to nalidixic acid, 21 (68%) to ciprofloxacin, 17 (54%) to cefotaxime and ceftazidime, and 17 (55%) to cefepime. For the others antibiotics included in the study, the resistance of the isolated strains was below 50%. All strains were susceptible to ceftazidime/clavulanic acid and cefotaxime/clavulanic acid, indicating that the resistance was caused by β-lactamase enzymes and not by other mechanisms, such as porin loss, or efflux pumps (Table 2).

Out of the total isolated strains (n = 31) from turkeys, six (19%) were classified as MDR (multidrug-resistant), being resistant to more than one antimicrobial substance from three different antibiotic classes, 23 (74%) were non-susceptible to at least one agent in all but two or fewer antimicrobial categories and were considered XDR (extensively drug-resistant), while 2 (7%) strains were susceptible to all antibiotic categories included in the study. The MARI (Multiple Antibiotic Resistance Index) considered only the strains resistant to at least one antibiotic class, with values ranging between 0.05 and 0.62. Overall, the MARI values for commensal *Escherichia coli* strains ranged from 0.10 to 0.71, while for ESBL/*Amp*C-producing strains, the values varied between 0.05 and 0.62. Among commensal strains isolated from turkeys, the MARI ranged from 0.29 to 0.62, whereas for ESBL/*Amp*C producers, it ranged from 0.05 to 0.57. For commensal *E. coli* strains isolated from broiler chickens, MARI values ranged from 0.10 to 0.71, and from 0.10 to 0.60 for ESBL/*Amp*C-producing strains. These findings indicate a higher rate of antibiotic resistance among isolates from broiler chickens, suggesting greater exposure to or more frequent use of antibiotics in broiler production systems.

Relatively, the isolates from broiler chickens showed the highest frequency of resistance to nalidixic acid (85%), ampicillin and ciprofloxacin (73%), cefotaxime (61%), ceftazidime (59%), and cefepime (56%). Some isolated strains from broilers also showed resistance to antibiotics from the cephalosporin class with clavulanic acid, in contrast to the bacteria identified from turkeys. Thus, specifically in broiler chickens, 10 strains (14%) were resistant to cefotaxime/clavulanic acid, and 11 strains (15%) were resistant to ceftazidime/clavulanic acid (Table 3).

The resistance to colistin was observed only in turkey isolates and with a frequency of 1%. Polymyxins, particularly colistin, were historically used as growth promoters in the livestock sector, especially in poultry. The ban on the use of polymyxins in animal feed has led to a reduction in antibiotic resistance of *E. coli* strains isolated from animals over several years. In this regard, colistin represents a conclusive example of the success of the “One Health” approach [32,33].

Statistical comparisons using chi-square test with Yates correction or Fisher’s exact test revealed significant differences (*p* < 0.05) in resistance rates between broiler and turkey isolates for ampicillin (AMP) and tetracycline (TET). No statistically significant differences were observed for the remaining antibiotics listed in Table 4.

Classification into resistance categories revealed that in broilers, 39 (55%) of the isolated strains were MDR, 28 (39%) were XDR, while 4 (6%) were susceptible to all categories of tested antibiotics.

The MARI ranged from 0.05 to 0.62. None of the isolated strains could be classified as PDR (Pan-Drug Resistant) because all isolated strains showed 100% susceptibility to certain antibiotics from the aminoglycoside category (amikacin), carbapenems (imipenem), and the tetracycline category (tigecycline). In total, six strains were susceptible to all categories of antibiotics (two strains isolated from turkeys and four from broiler chickens).

A classication of 102 *E. coli* isolates, based on their antimicrobial resistance profiles showed that 50% were extensively drug resistant (XDR)—comprising 44% ESBL producing and 6% commensal strains, while 44% were multidrug-resistant (MDR), including 15% ESBL and 29% commensal isolates (Figure 1). This distribution highlights the high burden of resistance among both commensal and ESBL-producing *E. coli* strains. To further characterize these patterns, a correlation and clustering analysis to explore potential relationships between reistance phenotypes was performed.

### 2.2. Correlation and Clustering Analysis of Antibiotic Resistance Patterns

To explore the potential co-resistance between antibiotics, a Pearson correlation matrix was generated using binary resistance data (R = 1, S = 0) across all tested isolates (Figure 2). The analysis revealed strong positive correlations between several β-lactam antibiotics, that belong to the cephalosporin subclass, notably cefotaxime (CTX), ceftazidime (CAZ), and cefepime (FEP) (r > 0.80), suggesting the presence of shared resistance mechanisms, such as ESBL production. Strong positive correlations (r > 0.80) were not limited to β-lactams such as CTX, CAZ, and FEP, but also extended to other antibiotics, including combinations like cefepime–foxitin (FEP–FOX) and cefepime–sulfamethoxazole (FEP–SMZ), reflecting broader co-resistance trends among the studied isolates. Moderate correlations were also observed between tetracycline (TET) and chloramphenicol (CHL), indicating possible cross-resistance or co-selection under similar antimicrobial pressure. Antibiotics such as amikacin (AMK) and colistin (COL) exhibited low correlation values with most other agents, while tigecycline (TIG) and azithromycin (AZI) showed moderate correlations with selected antibiotics, such as tetracycline (TET) and cefoxitin (FOX), potentially reflecting overlapping or emerging resistance trends. These findings suggest that while amikacin and colistin may remain largely unaffected by common cross-resistance mechanisms, other agents such as tigecycline and azithromycin may share partial resistance pathways with drugs like tetracycline and cefoxitin. Overall, these patterns highlight the value of correlation analysis of resistance profiles in identifying antibiotic classes that are likely to be compromised in multidrug-resistant (MDR) and extensively drug-resistant (XDR) bacterial populations. These correlation patterns are illustrated in Figure 2, where stronger associations between antibiotic resistance profiles are represented by warmer colors (e.g., red), and weaker or no correlations appear in cooler shades (e.g., blue to white).

To further characterize resistance patterns, a hierarchical clustering analysis was performed using Ward’s method on the same binary resistance dataset to identify patterns of similarity in resistance profiles among antibiotics and bacterial isolates. Using Ward’s method on binary resistance data (R/S) across 102 isolates, a dendrogram revealed distinct groupings of antibiotics with shared resistance behavior (Figure 3). Notably, β-lactam antibiotics such as cefotaxime, ceftazidime, and cefepime clustered closely together, consistent with their common mechanism of action and susceptibility to ESBL activity. Other antibiotics, such as tetracycline, chloramphenicol, and nalidixic acid formed a separate cluster, potentially reflecting frequent co-resistance in isolates from broiler chickens. In contrast, amikacin, imipenem, and tigecycline remained isolated, indicating minimal resistance and a distinct resistance profile. Clustering of isolates also revealed subgroups with highly similar multidrug resistance patterns, regardless of host origin, which may reflect clonal dissemination or shared selective pressures. This stratification highlights the value of clustering techniques in understanding resistance epidemiology and guiding antimicrobial stewardship.

### 2.3. Prevalence and Penetrance of Resistance Genes, Diagnostic Odds Ratio

In Table 5 are presented the distribution of strains into categories, either resistant (R) or susceptible (S) to the mentioned antibiotics, based on the results obtained from AST, as well as the prevalence of resistance genes: *Amp*C, *bla*Z, and *bla*TEM. Furthermore, the penetrance of the genes P (%) and the diagnostic odds ratio (DOR), were also calculated.

#### 2.3.1. Penetrance

The penetrance is expressed as a percentage and is defined as the proportion of individuals (in this case, strains) that carry a particular gene (*amp*C, *bla*Z or *bla*TEM) and exhibit the expected phenotype (antibiotic resistance to certain antibiotics) [24,34,35,36].

The highest penetrance values were obtained for the all three genes for ampicillin, respectively, *bla*TEM gene (80% for ampicillin), *Amp*C gene (79% for ampicillin), and *bla*Z gene (72.7% for ampicillin). In the case of *Amp*C gene higher penetrance values were also obtained for cefotaxime (62.5%), ceftazidime (58%) and cefepime (53%), while in *bla*TEM gene higher values of penetrance were observed for cefepime (63%), cefotaxime (58%) and ceftazidime (55%).

For all other antibiotics included in the study, the penetrance values for these genes were below 50%.

By calculating the penetrance for each gene individually, the following values are obtained:Penetrance *Amp*C = [246/(246 + 310)] × 100 = 44.2%(1)Penetrance *bla*Z = [40/(40 + 15)] × 100 = 72.7%(2)Penetrance *bla*TEM = [171/(171 + 123)] × 100 = 58.1%(3)

And total penetrance:P% = [468/(468 + 448)] × 100 = 51.09%(4)

The total penetrance value of 51.09% indicates that a large number of the isolated *E. coli* strains carry genes that induce antibiotic resistance to β-lactams, cephalosporins, or penicillins. However, the presence of these genes is not always correlated with their phenotypic expression.

A comprehensive evaluation of the genetic determinants of antibiotic resistance should be based on identifying genetic factors at the chromosomal level, not just on antimicrobial susceptibility testing profiles. Identifying these factors could play an important role in elucidating the processes that induce or, on the contrary, undermine the emergence of bacterial antibiotic resistance.

Penetrance depends on several factors, including the environment, epigenetic modifiers, modifier genes, and various intrinsic factors. Understanding these factors and how they influence the gene penetrance could help explain why, in some Gram-negative bacterial strains, these genes are phenotypically expressed, while in others, they are not.

#### 2.3.2. Diagnostic Odds Ratio

In this study, one factor was antimicrobial resistance, while the other was the presence of resistance genes (Table 4). Essentially, the aim was to determine the extent to which the occurrence of phenotypic resistance is correlated with the presence of genotypic resistance by calculating the odds ratio for the antibiotics included in the study.

The results presented in Table 4 showed that the DOR values ranged from 0.67 to 81. These values can be used to estimate how AST using the microdilution method can indicate the presence of antibiotic resistance genes in the tested strains.

In the case of the *AmpC* gene, the DOR values obtained were 3.35 for ampicillin, 0.67 for cefepime, 3.99 for cefoxitin, 2.13 for cefotaxime with clavulanic acid, and 2.04 for ceftazidime with clavulanic acid. On the other hand, for the *bla*TEM gene, the DOR values were 1.62 for ampicillin, 0.91 for cefotaxime, and 3.26 for cefoxitin.

However, for some antibiotics, subunitary DOR values were obtained, as in the case of the *AmpC* gene with cefepime (DOR = 0.67), the *bla*Z gene for ampicillin (DOR = 0.72) or the *bla*TEM gene with cefotaxime (DOR = 0.91) and ceftazidime (DOR = 0.70).

Penetrance (P%) and diganostic odds ratio (DOR) are two extremely valuable indicators that take into account both genotypic and phenotypic resistance and determine the effectiveness of a test used to evaluate the phenotypic resistance to antimicrobials.

Diagnostic odds ratio values greater than 1 indicate that the AST method used is effective in indicating the presence of antibiotic resistance genes in the studied isolates.

## 3. Discussion

The present study reports the antimicrobial resistance profile of 60 ESBL-producing *E. coli* strains isolated from fecal samples of broilers (n = 43) and turkeys (n = 17) from the North-Western part of Romania. The genotypic antimicrobial resistance characteristics of the isolates were investigated by assessing the occurrence of some resistance genes (*Amp*C, *bla*Z and *bla*TEM) corresponding to β-lactams (seven antibiotics) and carbapenems (one antibiotic) classes of antimicrobials. The findings of this study highlight the persistent challenge of antimicrobial resistance in *E. coli* strains sourced from broiler chickens and fattening turkeys. The isolation of presumptive ESBL/*Amp*C-producing strains at high levels (61% in broiler chickens and 55% in turkeys) reflects a significant public health concern and highlights the roles that agricultural practices play in the dissemination of antibiotic resistance. The results corroborate previous reports of *E. coli* resistance trends in poultry, but also livestock and wild birds [13,14,15,37,38]. For instance, Athanasakopoulou et al. (2021) [13] reported ESBL/*Amp*C-producing *E. coli* in 100% of swine and cattle isolates and 1.2% in wild birds, while Tofani et al. (2022) [19] detected ESBL/*Amp*C-producing *E. coli* in 156 out of 809 broiler cecal samples, with 84% phenotypically classified as ESBL producers and 100% being MDR. Similarly, Zahra et al. (2024) [20] found 63% *E. coli* positivity in broiler intestinal samples from Indonesia, with 22.2% being MDR and 6% confirmed as ESBL producers.

The complex characteristics of bacterial resistance, especially in *E. coli*, can primarily be linked to the enzyme-driven inactivation of β-lactam antibiotics, particularly through the production of ESBL and *Amp*C β-lactamases [14,16]. Notably, the dominant presence of the *Amp*C gene, as well as *bla*TEM, *bla*Z genes, and various ESBL variants echoes findings in other research studies that emphasize the need for careful monitoring of resistance patterns [14,18,26]. In support, a study conducted on broiler chickens, in Italy, reported the presence of *bla*CTX-M-1 gene in 42.7% of isolates (mainly CTX-M-15), *bla*TEM in 29%, and *bla*SHV in 19.8% [20]. In a different study from Greece, Athanasakopoulou et al. (2021) similarly observed 100% prevalence of *bla*CTX-M-1/15, with *bla*TEM found in 68.42% of ESBL *E. coli,* isolated from livestock [13]. The presence of ESBL genes was also detected in 99% of the ESBL-producing *E. coli* isolated from broilers in Brazil. Thus, *bla*CTX-M group 2 dominant (55%), followed by *bla*CTX-M group 1 (38%), and *bla*CTX-M group 8 (10%) [39]. In Indonesia, Zahra et al. (2024) noticed that four out of six ESBL-positive strains carried both *bla*TEM and *bla*CTX-M genes, while two isolates only carried the *bla*CTX-M gene [20]. These reports affirm the widespread distribution of these β-lactamase genes. The penetrance of the *Amp*C gene (76.2%) and the *bla*Z (72.7%) gene in the isolates suggests that these resistance mechanisms are well established within the bacterial populations studied. As comparable data on *Amp*C gene, other studies have similarly reported the presence of *bla*CIT, *bla*FOX, *bla*CMY, *bla*CMY-2, *bla*TEM-52c, *bla*SHV-12, and the *Amp*C-type gene *cit*, often in combination with *bla*CTX-M. Although the *bla*Z gene was not explicitly identified, the detection of related β-lactamase genes in poultry production systems reinforces the relevance of our findings [14,18,19,21,39].

Interestingly, while there has been a slight reduction in the prevalence of ESBL/*Amp*C strains in the EU as mentioned in reports from 2022, such trends must be interpreted with caution. With resistance levels still exceedingly high-evident from the high rates of extensively drug-resistant (XDR) and multidrug resistant (MDR) phenotypes observed in the isolates, it is imperative to continue evaluating the bacteriological environment of poultry and its implications on human health [23,33,35]. This is supported by findings from other researchers, where 100% of their ESBL-producing *E.coli* isolates were classified as MDR [13,19,40].

The variability observed in the occurrence of *E. coli* strains producing the ESBL/*Amp*C phenotype across different EU countries [13,19,21,41,42,43] reflects an alarming range of antimicrobial pressure and interventions within poultry farming systems. Such disparities emphasize the impact of national policies and antibiotic stewardship practices on microbial resistance profiles. The somewhat improved situation in certain EU countries could be attributed to stringent regulations against the irrational use of antibiotics, a measure that needs to be uniformly adopted across all regions to better combat antimicrobial resistance, as reinforced by a studies which highlight how differences in slaughterhouse hygiene, environment and batch contamination influenced ESBL and *Amp*C beta-lactamase producing *Enterobacteriaceae* prevalence in broiler chickens, advocating for policy-level interventions at both farm and processing levels [14,43,44].

The findings indicate that while broiler chickens displayed a higher percentage of resistant isolates overall, turkeys exhibited unexpected levels of resistance for specific antibiotics such as ampicillin and gentamicin. This illustrates the complexity of resistance development and the need for site-specific interventions. The escalating resistance rates, particularly to cefotaxime/clavulanic acid and ceftazidime/clavulanic acid, pose substantial challenges in treating infections, due to the broad implications of β-lactam antibiotic failures. Similar resistance patterns were observed by other researchers in broiler chickens, where elevated resistance rates to ampicillin, cephalosporins from the first-generation (cefazolin, cephalexin), third-generation (cefotaxime, cefoperazone) and fourth generation (cefepime, cefquinome), trimethoprim-sulfamethoxazole, fluoroquinolones (ciprofloxacin, flumequine, enrofloxacin), aminoglycosides (gentamicin, streptomycin), monobactam (aztreonam) and tetracycline, with production systems (conventional vs. organic) influencing susceptibility profiles [13,19,20,39,41,45].

Furthermore, the significant correlation identified between the phenotypic resistance and the presence of specific resistance genes bolsters the conclusion that careful screenings and genetic assessments should be part of monitoring strategies for AMR. The diagnostic odds ratios calculated in this study further reinforce the reliability of the antimicrobial susceptibility testing methods employed and affirm the consistency of the phenotypic data with underlying genetic determinants. The utility of integrating molecular diagnostics into surveillance systems is also confirmed by other researchers, who emphasized strong genotype-phenotype correlations in E. coli resistance, with consistent associations between the presence of *bla*CTX-M, *bla*TEM, or *fos*A3 and corresponding phenotypic resistance [19,39].

Overall, this discussion reflects the need for continued collaboration between public health entities and agricultural sectors to address and mitigate antimicrobial resistance, particularly in poultry sector. The trends observed in present study call for an urgent need for enhanced surveillance, bolstered antibiotic stewardship, and research into alternative therapeutics and management practices. Future investigations should not only explore the molecular mechanisms that may contribute to increased resistance but also consider environmental and animal welfare impacts across the agricultural framework. The health of both animal populations and humans are indivisible linked—a reality that mandates a sustainable, integrated approach to tackling multidrug resistance and its consequences in *E. coli* strains derived from poultry.

## 4. Materials and Methods

### 4.1. Material Selection

From January 2020 to December 2022, a total of 102 fecal samples from poultry (turkeys and broiler chickens) from various farms in northwest Romania were examined in this study. The samples came from local slaughterhouses that provided services to these farms. An ante-mortem examination performed by licensed veterinary professionals before slaughter revealed that every animal was clinically healthy.

A total of 31 samples were collected from turkeys aged 13 to 21 weeks, while 71 samples were collected from broiler chickens aged 36 to 50 days. The exact slaughter age of each lot varied according to the farm of origin’s management practices.

The birds were raised on commercial farms, but detailed information on specific housing systems, feeding regimens, and environmental controls was not available for all of the flocks studied. However, it is common for poultry farms in this region to use standardized intensive farming practices. All animals were raised specifically for commercial meat production.

During the ante-mortem inspection, no signs of disease were detected. During the sampling period, no acute clinical outbreaks were reported to the slaughterhouses, despite the fact that thorough veterinary records were not collected from every farm.

Cecums were harvested immediately after slaughter and refrigerated before being delivered to the laboratory the same day. All samples were processed within 24 h of arrival, and inoculation was performed using cecal contents.

### 4.2. Isolation of E. coli Strains

The samples were processed for the isolation of *Escherichia coli* strains, using buffered peptone water (Sigma-Aldrich, Darmstadt, Germany) as an enrichment medium. One gram of feces was inoculated into 9 mL of buffered peptone water, and incubation was carried out at 37 °C for 20 h.

For the isolation of presumptive ESBL/*Amp*C/carbapenemase-producing strains, two selective solid media were used: MacConkey agar (Sigma-Aldrich, Darmstadt, Germany) and MacConkey agar supplemented with cefotaxime (1 mg/L, Sigma-Aldrich, Germany). A volume of 10 μL from the enriched culture was inoculated onto the surface of both types of selective solid media [29,30,31]. Incubation was carried out at 44 °C for 20 h. The isolated colonies were morphologically analyzed, followed by biochemical testing using Api 20E system (BioMérieux, Lyon, France). Testing was performed on a single isolated colony for each of the 102 samples to ensure the processing of a pure *E. coli* strain.

Strains that grew on the cefotaxime-supplemented medium were considered presumptive producers of ESBL, *Amp*C, and/or carbapenemases. The isolated strains were suspended in 2 mL of 10% glycerol and stored at −80 °C for PCR analyses.

### 4.3. Antimicrobial Susceptibility Testing (AST) Using the Microdilution Plate Method

From the 24 h primary culture obtained on agar, 3–4 colonies were transferred into 5 mL of distilled water, and turbidity was adjusted to 0.5 McFarland. From the obtained inoculum, 10 μL was transferred into cation-adjusted Mueller-Hinton broth (Thermo Scientific™, Dreieich, Germany), and the antibiotic susceptibility testing (AST) plate was inoculated, within a maximum of 30 min. For each well containing 50 μL of antimicrobial agent diluted in broth, 50 μL of bacterial suspension was added. The inoculated plates were sealed with adhesive film to prevent drying and incubated at 35 °C for 20 h [29].

The antibiotic susceptibility panel included antibiotics from the following classes: Aminoglycosides (gentamicin-GEN, amikacin-AMK), Amphenicols (chloramphenicol-CHL), β-lactams (ampicillin-AMP, cefotaxime-CTX, ceftazidime-CAZ, cefoxitin-FOX, cefepime-FEP), β-lactams with clavulanic acid (cefotaxime/clavulanic acid-CTX/CA, ceftazidime/clavulanic acid-CAZ/CA), Quinolones (ciprofloxacin-CIP, nalidixic acid-NAL), Folate inhibitors (sulfamethoxazole-SMZ, trimethoprim-TMP), Carbapenems (meropenem-MRP, imipenem-IMP, ertapenem-ETP), Polimyxins (Colistin-COL), Tetracyclines (tetracycline-TET, tigecycline-TIG) and Macrolides (azithromycin-AZI).

Plate readings were performed using the Biomic V3 analyzer (Giles Scientific Inc., San Diego, CA, USA). Based on the obtained minimum inhibitory concentrations (MICs), the strains were classified as resistant (R) or susceptible (S).

### 4.4. DNA Extraction and Gene Detection by PCR

The extraction was performed using the InstaGENE (Bio-Rad, Hercules, CA, USA) matrix, following the manufacturer’s instructions. An isolated colony was taken and suspended in 1 mL of distilled water, then centrifuged at 10,000–12,000 rpm for 1 min. The supernatant was removed. To the obtained pellet, 200 μL of InstaGENE (Bio-Rad, USA) matrix was added, and the sample was incubated for 15–30 min at 56 °C. It was vortexed for 10 min, then placed in a heating block at 100 °C for 8 min. This was followed by another vortexing step for 5 min and a centrifugation step for 2–3 min at 10,000–20,000 rpm. From the obtained supernatant, 25 μL was used for a 50 μL PCR reaction.

Amplification was performed using the Eppendorf Mastercycler PRO S 6325 Thermal Cycler (22331 Hamburg, Germany), in a 25 μL reaction mix containing 2.5 μL Dream Taq Buffer, 0.5 μL MgCl_2_, 1 μL dNTP, 0.3 μL Taq DNA Polymerase, 0.5 μL of each primer (Table 6), 3 μL of extracted DNA, and PCR-grade water. The thermal cycling conditions included an initial denaturation and polymerase activation step at 95 °C for 2 min, followed by 32 cycles of denaturation at 95 °C for 30 s, primer alignment at 60 °C for 30 s, and extension at 72 °C for 30 s. Cooling was performed at 4 °C for 5 min. Electrophoresis was carried out on an agarose gel with migration at 100 V, 150 mA, for 40–60 min (Appendix A) [46,47,48].

### 4.5. Statistical Analysis of Phenotypic and Genotypic Antibiotic Resistance

Based on the results obtained from phenotypic antibiotic resistance testing by AST, the isolated strains were phenotypically classified into the following categories:

MDR (multidrug resistance):Multidrug resistance;Non-susceptible to >1 agent from ≥3 classes of antibiotics.

XDR (extensively drug resistance):Extended antibiotic resistance;Non-susceptible to >1 agent from all categories of antibiotics except for ≤2, or susceptible only to ≤2 of the antibiotic categories.

PDR (pan-drug resistance):pan-antibiotic resistance;Non-susceptible to all categories of antibiotics tested [49,50,51].

The Multiple Antibiotic Resistance Index (MARI) was calculated according to Krumperman’s formula:MARI = a/b(5)
where “a” represents the number of antibiotics to which the isolated strain showed resistance, and “b” represents the total number of antibiotics for which the respective strain was tested.

To estimate the extent to which bacteria carrying resistance genes express these genes phenotypically, the penetrance (P%) and the diagnostic odds ratio were calculated.

Based on results, the isolated strains were divided into four categories:RG+ = phenotypically resistant strains that carry antibiotic resistance genes;RG− = phenotypically resistant strains that do not carry antibiotic resistance genes;SG+ = phenotypically susceptible strains that carry the genes;SG− = phenotypically susceptible strains that do not carry the genes.

The penetrance, expressed as a percentage, represents the ratio between the number of strains that show the phenotype and also carry the genes, and the total strains that have the genotype, as follows:Penetrance (%) = [(RG^+^)/(RG^+^ + SG^+^)] × 100(6)

Subsequently, the diagnostic odds ratio (DOR) was calculated to evaluate the association between genotypic resistance and the phenotypic resistance obtained through AST.DOR = (RG^+^ × SG^−^)/(RG^−^ × SG^+^)(7)

The graphics were obtained using the Python 3.11.8 (Python Software Foundation, Wilmington, DE, USA) program.

Statistical analysis was performed using IBM SPSS Statistics v26.0 (IBM Corp., Armonk, NY, USA). To evaluate the correlation between poultry type (broilers vs. turkeys) and the type of *E. coli* strain (commensal vs. ESBL/*Amp*C-producing), a chi-square test of independence was applied to the data in Table 1. A *p*-value < 0.05 was considered statistically significant.

For Table 4, the chi-square test with Yates correction or Fisher’s exact test was used to assess whether the observed differences in antibiotic resistance and susceptibility frequencies between turkeys and broilers were statistically significant (*p* < 0.05).

In addition, diagnostic odds ratios (DOR) were calculated to estimate the strength of association between phenotypic resistance and the presence of resistance genes. Penetrance (P%) was also computed to express the proportion of resistant strains carrying a corresponding gene. These calculations were performed using standard 2 × 2 contingency table formulas.

## 5. Conclusions

The proportion of presumptive ESBL/*Amp*C-producing strains isolated was 55% in turkeys and 61% in broiler chickens. Antimicrobial resistance was high, with 50% of the total isolated strains categorized as XDR, 44% as MDR, while only 6% showed fully susceptibility to all antibiotics.

The percentage of isolates resistant to the antibiotics studied was generally higher in broiler chickens compared to those isolated from turkeys, except for ampicillin and gentamicin, where resistance was higher. Some isolates from broilers showed resistance to cefotaxime/clavulanic acid (15%), respectively, to ceftazidime/clavulanic acid (14%).

Penetrance values for the three genes studied were 44.2% for the *Amp*C gene, 72.7% for the *bla*Z gene, and 58.1% for the *bla*TEM gene. The total penetrance was 51.09%, while the DOR values ranged from 0.67 to 81. All these values indicate that the AST method used in this study effectively indicates the presence of antibiotic resistance genes in the isolates. Future AMR surveillance must prioritize molecular diagnostics and environmental monitoring, alongside traditional AST, to support a sustainable One Health strategy.

## 6. Study Limitations

The presented study has some limitations that should be mentioned. First of all, the research was carried out within a limited geographical area (northwestern part of Romania), which may restrict the applicability of the results to wider poultry populations with different epidemiological chararcteristics. Also, the low number of isolated and analyzed strains may represent a limitation. Secondly, even if the antibiotic resistance was searched for several classes of antibiotics, the PCR was performed with a restricted gene panel, respectively, for only three genes. Finally, there is a lack of whole genome sequencing determination, a method which provides comprehensive information about the genetic structure of the bacterium, including both core genes and accessory genes, which may comprise resistance genes.

## Figures and Tables

**Figure 1 antibiotics-14-00578-f001:**
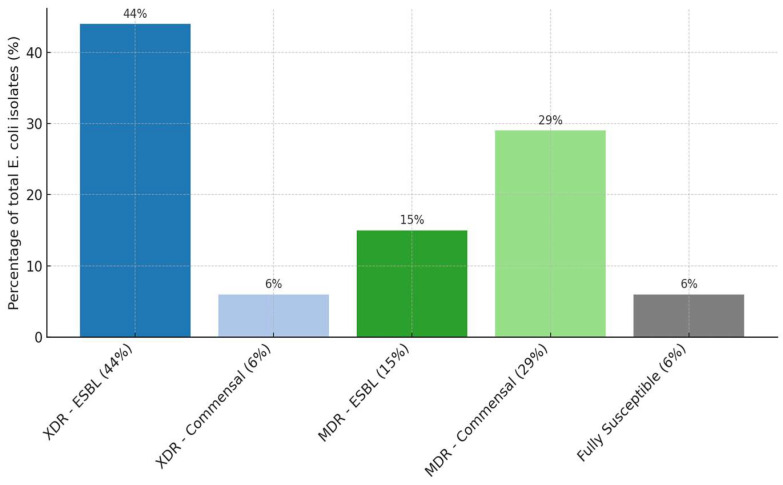
*E. coli* isolates: MDR, XDR and 100% susceptible strains. Legend: XDR = extensively drug resistant isolates; MDR = multidrug resistant isolates; ESBL = extended-spectrum beta-lactamase producers isolates; commensal = non-ESBL/*Amp*C isolates with antibiotic resistance; fully susceptible = strains not resistant to any tested antibiotics.

**Figure 2 antibiotics-14-00578-f002:**
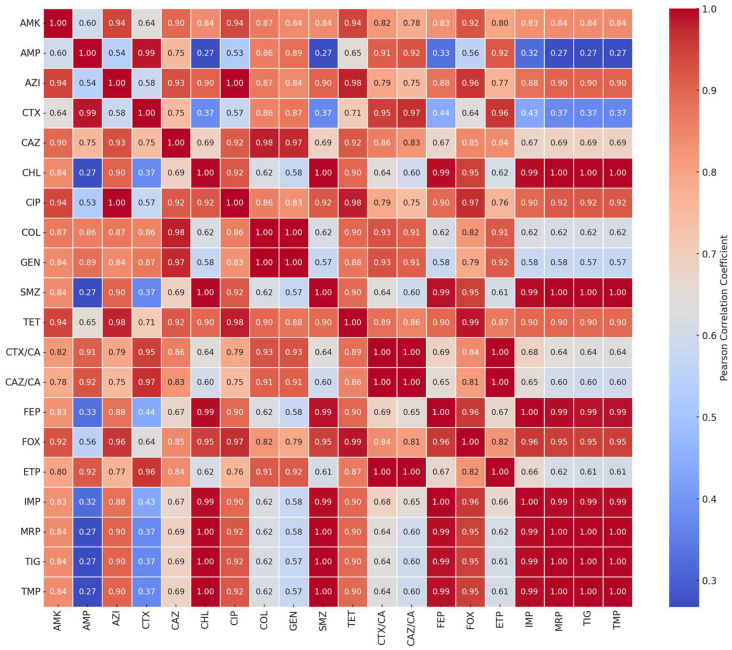
Correlation analysis of antibiotic resistance patterns. Legend: red = strong correlation, blue, = weak correlation, AMP (ampicillin), CHL (chloramphenicol), GEN (gentamicin), AMK (amikacin), CTX (cefotaxime), CAZ (ceftazidime), FOX (cefoxitin), FEP (cefepime), CTX/CA (cefotaxime with clavulanic acid), CAZ/CA (ceftazidime with clavulanic acid), CIP (ciprofloxacin), NAL (nalidixic acid), SMZ (sulfamethoxazole), TMP (trimethoprim), MRP (meropenem), IMP (imipenem), ETP (ertapenem), COL (colistin), TET (tetracycline), TIG (tigecycline), and AZI (azithromycin).

**Figure 3 antibiotics-14-00578-f003:**
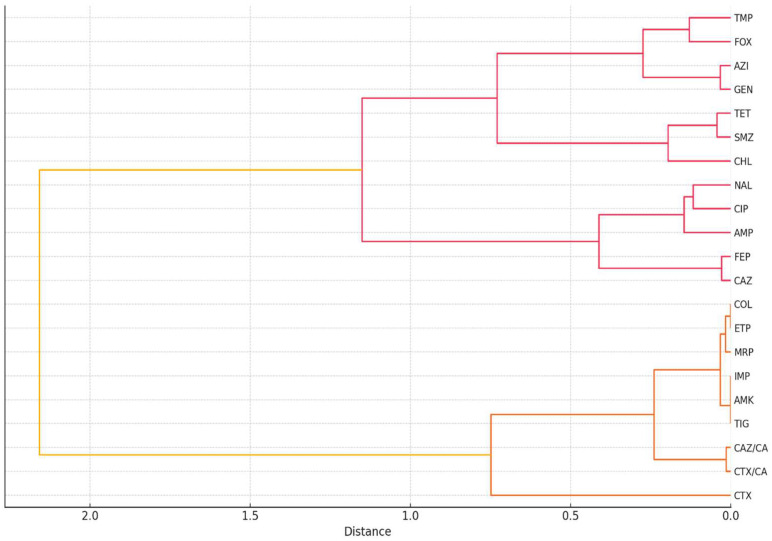
Clustering of antibiotics based on resistance profiles (%). Legend: AMP (ampicillin), CHL (chloramphenicol), GEN (gentamicin), AMK (amikacin), CTX (cefotaxime), CAZ (ceftazidime), FOX (cefoxitin), FEP (cefepime), CTX/CA (cefotaxime with clavulanic acid), CAZ/CA (ceftazidime with clavulanic acid), CIP (ciprofloxacin), NAL (nalidixic acid), SMZ (sulfamethoxazole), TMP (trimethoprim), MRP (meropenem), IMP (imipenem), ETP (ertapenem), COL (colistin), TET (tetracycline), TIG (tigecycline), and AZI (azithromycin). Red branches indicate tightly clustered antibiotics with high similarity (low linkage distances), orange branches represent intermediate-level clusters formed by merging smaller groups, and yellow branches denote the highest-level clusters encompassing the most diverse antibiotic groupings.

**Table 1 antibiotics-14-00578-t001:** The total number of commensal and presumptive ESBL/*Amp*C-producing isolated strains.

Species	Commensal Strains No. (%)	Presumptive ESBL/*Amp*C Strains No. (%)	Total	
Turkeys	14 (45)	17 (55)	31	X^2^ = 0.1 *p* = 0.75
Broiler chickens	28 (39)	43 (61)	71	
Total	42 (41)	60 (59)	102	

Legend: ESBL—broad-spectrum β-lactamases.

**Table 2 antibiotics-14-00578-t002:** Antimicrobial resistance behavior of the *E. coli* isolated from turkeys.

Antibiotic Class	Antibiotic	Cut-Off Value (µg/L)	MIC (µg/L)	Resistant (No./%)	Susceptibile (No./%)
Aminoglycosides	GEN	8	0.5–≥16	13 (42)	18 (58)
	AMK	8	≤4	0 (0)	31 (100)
Amphenicols	CHL	16	0.8–≥64	23 (74)	8 (26)
β-lactams (penicillins, cephalosporins, 1st, 2nd and 3rd generation cephalosporins)	AMP	16	1–≥32	25 (81)	6 (19)
	CTX	0.25	0.25–≥4	17 (54)	14 (45)
	CAZ	0.5	0.25–≥8	17 (54)	14 (45)
	FOX	8	0.5–≥64	9 (29)	22 (71)
	FEP	0.125	0.064–≥32	17 (55)	14 (45)
	CTX/CA	0.25	0.064–64	0 (0)	31 (100)
	CAZ/CA	0.5	0.125–128	0 (0)	31 (100)
Quinolones	CIP	0.064	0.015–8	21 (68)	10 (32)
	NAL	8	4–≥64	22 (71)	9 (29)
Folate inhibitors	SMZ	64	8–512	23 (74)	8 (26)
	TMP	2	0.5–≥16	13 (42)	18 (58)
Carbapenems	MRP	0.125	0.03–0.125	0 (0)	31 (100)
	IMP	0.5	≤0.125	0 (0)	31 (100)
	ETP	0.064	0.016–0.064	0 (0)	31 (100)
Polimyxins	COL	2	1–2	0 (0)	31 (100)
Tetracyclines	TET	8	2–≥64	24 (77)	7 (23)
	TIG	0.5	≤0.25	0 (0)	31 (100)
Macrolides	AZI	16	2–≥64	14 (45)	17 (55)

Legend: AMP (ampicillin), CHL (chloramphenicol), GEN (gentamicin), AMK (amikacin), CTX (cefotaxime), CAZ (ceftazidime), FOX (cefoxitin), FEP (cefepime), CTX/CA (cefotaxime with clavulanic acid), CAZ/CA (ceftazidime with clavulanic acid), CIP (ciprofloxacin), NAL (nalidixic acid), SMZ (sulfamethoxazole), TMP (trimethoprim), MRP (meropenem), IMP (imipenem), ETP (ertapenem), COL (colistin), TET (tetracycline), TIG (tigecycline), and AZI (azithromycin). The strains with high to very high level of resistance were highlighted for emphasis.

**Table 3 antibiotics-14-00578-t003:** Antimicrobial resistance behavior of the *E. coli* isolated from broilers.

Antibiotic Class	Antibiotic	Cut-Off Value (µg/L)	MIC (µg/L)	Resistant (No./%)	Susceptibile (No./%)
Aminoglycosides	GEN	8	0.5–≥16	5 (7)	66 (93)
	AMK	8	≤4	0 (0)	71 (100)
Amphenicols	CHL	16	0.8–≥64	15 (21)	56 (79)
β-lactams (penicillins, cephalosporins, 1st, 2nd and 3rd generation cephalosporins)	AMP	16	1–≥32	52 (73)	19 (27)
	CTX	0.25	0.25–≥4	43 (61)	28 (39)
	CAZ	0.5	0.25–≥8	42 (59)	29 (41)
	FOX	8	0.5–≥64	17 (24)	54 (76)
	FEP	0.125	0.064–≥32	40 (56)	31 (44)
	CTX/CA	0.25	0.064–64	11 (15)	60 (85)
	CAZ/CA	0.5	0.125–128	10 (14)	61 (86)
Quinolones	CIP	0.064	0.015–8	52 (73)	19 (27)
	NAL	8	4–≥64	60 (85)	11 (15)
Folate inhibitors	SMZ	64	8–512	26 (37)	45 (63)
	TMP	2	0.5–≥16	18 (25)	53 (75)
Carbapenems	MRP	0.125	0.03–0.125	2 (3)	69 (97)
	IMP	0.5	≤0.125	0 (0)	71 (100)
	ETP	0.064	0.016–0.064	1 (1)	70 (99)
Polimyxins	COL	2	1–2	1 (1)	70 (99)
Tetracyclines	TET	8	2–≥64	28 (39)	43 (61)
	TIG	0.5	≤0.25	0 (0)	71 (100)
Macrolides	AZI	16	2–≥64	5 (7)	66 (93)

Legend: AMP (ampicillin), CHL (chloramphenicol), GEN (gentamicin), AMK (amikacin), CTX (cefotaxime), CAZ (ceftazidime), FOX (cefoxitin), FEP (cefepime), CTX/CA (cefotaxime with clavulanic acid), CAZ/CA (ceftazidime with clavulanic acid), CIP (ciprofloxacin), NAL (nalidixic acid), SMZ (sulfamethoxazole), TMP (trimethoprim), MRP (meropenem), IMP (imipenem), ETP (ertapenem), COL (colistin), TET (tetracycline), TIG (tigecycline), and AZI (azithromycin). The strains with high to very high level of resistance were highlighted for emphasis.

**Table 4 antibiotics-14-00578-t004:** Comparison between antimicrobial resistance patterns of *E. coli* strains isolated from broilers and turkeys.

Crt. No.	Antibiotic	Broilers		Turkeys		X^2^ Value	*p*-Value
		Resistant (No)	Susceptible (No)	Resistant (No)	Susceptible (No)		
1.	AMP	48	23	29	42	9.19	0.0024
2.	CTX	33	38	27	44	0.72	0.3957
3.	CAZ	32	39	27	44	0.46	0.4959
4.	FEP	32	39	25	46	1.06	0.3043
5.	FOX	13	58	13	55	0.01	0.9203
6.	CHL	13	58	23	45	3.58	0.0522
7.	SMZ	26	45	23	45	0.03	0.8624
8.	TET	37	34	22	49	5.68	0.0171
9.	TMP	31	40	19	52	3.74	0.0531
10	NAL	10	61	9	62	0	1
11.	AMK	5	66	4	67	*	0.9999
12.	AZI	18	53	11	60	1.56	0.2116
13.	CAZ/CA	22	49	17	54	0.57	0.4502
14.	CTX/CA	23	48	22	49	0	1
15.	GEN	11	60	8	63	0.24	0.6242
16.	CIP	27	44	18	53	2.08	0.1492
17.	ETP	3	68	4	67	*	1
18.	IMP	2	69	3	68	*	1
19.	MRP	3	68	2	69	*	1
20.	TIG	2	69	1	70	*	1
21.	COL	6	65	3	68	*	0.4934

Legend: AMP (ampicillin), CHL (chloramphenicol), GEN (gentamicin), AMK (amikacin), CTX (cefotaxime), CAZ (ceftazidime), FOX (cefoxitin), FEP (cefepime), CTX/CA (cefotaxime with clavulanic acid), CAZ/CA (ceftazidime with clavulanic acid), CIP (ciprofloxacin), NAL (nalidixic acid), SMZ (sulfamethoxazole), TMP (trimethoprim), MRP (meropenem), IMP (imipenem), ETP (ertapenem), COL (colistin), TET (tetracycline), TIG (tigecycline), and AZI (azithromycin), * Fisher’s exact test.

**Table 5 antibiotics-14-00578-t005:** Distribution of isolates by categories, gene prevalence, penetrance and diagnostic odds ratio.

Gene	Antibiotic	R	S	RG+ nr. (%)	RG− nr. (%)	SG+ nr. (%)	SG− nr. (%)	P (%)	DOR
*amp*C	AMP	77	25	69 (90)	8 (10)	18 (72)	7 (28)	**79**	3.35
	CTX	60	42	50 (83)	10 (17)	30 (71)	12 (29)	62.5	2.00
	CAZ	59	43	45 (76)	14 (24)	32 (74)	11 (26)	58	1.10
	FEP	57	45	40 (70)	17 (30)	35 (78)	10 (22)	53	0.67
	FOX	26	76	23 (88)	3 (12)	50 (66)	26 (34)	31.5	**3.99**
	CTX AC	11	91	10 (91)	1 (9)	75 (82)	16 (17)	12	2.13
	CAZ AC	10	92	9 (90)	1 (10)	75 (82)	17 (18)	11	2.04
Subtotal		300	414	246	54	310	104		
*bla*Z	AMP	77	25	40 (52)	37 (48)	15 (60)	10 (40)	**72.7**	0.72
Subtotal		77	25	40	37	15	10		
*bla*TEM	AMP	77	25	40 (52)	37 (48)	10 (40)	15 (60)	**80**	1.62
	CTX	60	42	30 (50)	30 (50)	22 (52)	20 (48)	58	0.91
	CAZ	59	43	35 (59)	24 (41)	29 (67)	14 (33)	55	0.70
	FEP	57	45	34 (60)	23 (40)	20 (44)	25 (56)	63	1.85
	FOX	26	76	14 (54)	12 (46)	20 (26)	56 (74)	41	3.26
	CTX/AC	11	91	10 (91)	1 (9)	10 (11)	81 (89)	50	**81**
	CAZ/AC	10	92	8 (80)	2 (20)	12 (13)	80 (87)	40	**26.66**
	MRP	2	100	1 (50)	1 (50)	0	100 (100)	-	-
Subtotal		302	514	172	130	123	391		
TOTAL		679	953	468	211	448	505		

Legend: R—number of resistant isolates; S—number of susceptible isolates; RG+—resistant, gene-positive isolates; RG−—resistant, gene-negative isolates; SG+—susceptible, gene-positive isolates; SG−—susceptible, gene-negative isolates; *p* (%)—prevalence of gene-positive isolates; DOR—diagnostic odds ratio; AMP (ampicillin), CHL (chloramphenicol), GEN (gentamicin), AMK (amikacin), CTX (cefotaxime), CAZ (ceftazidime), FOX (cefoxitin), FEP (cefepime), CTX/CA (cefotaxime with clavulanic acid), CAZ/CA (ceftazidime with clavulanic acid), CIP (ciprofloxacin), NAL (nalidixic acid), SMZ (sulfamethoxazole), TMP (trimethoprim), MRP (meropenem), IMP (imipenem), ETP (ertapenem), COL (colistin), TET (tetracycline), TIG (tigecycline), and AZI (azithromycin). Higher *p* (%) and DOR values were highlighted in bold for emphasis.

**Table 6 antibiotics-14-00578-t006:** Primers used for the amplification of antibiotic resistance genes.

Gene	Primer Sequence	Elongation Temperature (°C)	Amplicons (pB)
*Amp*C	F: ATCAAAACTGGCAGCCG R: GAGCCCGTTTTATGCACCCA	65	510
*bla*Z	F: ACT TCA ACA CCT GCT GCT TTC R: TGA CCA CTT TTA TCA GCA ACC	60	490
*bla*TEM	F: GAGTATTCAACATTTCCGTGTC R: TAATCAGTGAGGCACCTATCTC	42	850

## Data Availability

All data generated or analyzed during this study are included in the submitted version of the manuscript.

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
