# Peer review of "Phenotypic and Genotypic Characterization of ESBL and *Amp*C β-Lactamase-Producing *E. coli* Isolates from Poultry in Northwestern Romania"

_antibiotics, 2025, doi:10.3390/antibiotics14060578_

Round 1
Reviewer 1 Report
Comments and Suggestions for Authors
Dear authors,
Please find the attached word file for the comments.

Reviewer 2 Report
Comments and Suggestions for Authors
Dear Authors, I have read your work thoroughly and have made comments and suggestions to help improve the body of the work. The article looks good and will benefit readers if the following are considered.
- In your abstract, typo errors- line 18- “ strins” should be checked. Also, Odds ratio ranges (0.67 to 3.99) are mentioned but not tied to specific comparisons. The Abstract needs a review to capture the important part of the result.
- In the introduction section, can the authors include more literature on poultry antibiotic trends and resistance, and include a more detailed justification of studying and detecting ESBL and AmpC 2 β-lactamase-Producing E. coli in poultry? Why choose the gene of interest, is it based on the prevalence from previous studies in same region/other region or because of its clinical relevance? What is the rationale for selecting the genes of interest? The study is interesting but requires detailed justification. There are some of statements in your introduction that require citations e.g line 54.
- In your methodology, you mentioned that the study was carried out from January 2020 to December 2022. This is 2025. I am curious to know what happened during 2023 to date. Was the study carried out and left unpublished? Were the isolates preserved? If the isolates were preserved, where/how?
- Also in your methodology, there was no mention of how the sample size were obtained. Was it randomly selected or convinence samples? How did you select the farms and 71 samples were collected to avoid bias. Can the Authors also clarify the ethical statement regarding animal sampling?
- In your antimicrobial analysis/testing, was it CLSI or EUCAST breakpoints that was used? Include the reference and standard. The media used should have their company and the country of manufacturing.
- The genes detection should be clarified before classifications of phenotypes and genotypes. How was the gene identified? Was gel electrophoresis used to confirm product size or what method was used to identify genes? What are the criteria for selecting only ampc, blaZ, and blaTem? What (ie. Negative or positive) control methods was used for each gene PCR?
- The Authors should consider a section in the methodology to discuss the method of “statistical analysis”, where the DOR and other statistical methods used can be discussed. I am also suggesting a chi-square on table 1 to determine a statistically significant association between poultry type and the strain type (Commensal vs ESBL/AmpC) with the p-value. It will give the work more value. Fishers test can also be carried out on table 2 and 3, for each the resistance and sensitivity frequency of the Antibiotic class or Antibiotic, respectively, to enhance comparison.
- Please confirm the labeling of your tables. Table 3 appeared twice.
- In the discussion sections, Authors should please avoid overstatement without referencing/citations.
- I am suggesting a section for “Limitation” where Authors can explain some of the limitations they faced during the research and gaps that need to be filled. Because determining only 3 genes can miss key ESBL variants.
Reviewer 3 Report
Comments and Suggestions for Authors
Dear colleagues!
I would like to note the undeniable importance of such studies.
But I have the following comments:
95-96 - “Out of the 102 fecal samples, 31 were collected from turkeys with a slaughter age 95 between 13 and 21 weeks, while 71 samples were collected from broiler chickens aged 96 between 36 and 50 days.“ - move to the “Materials and Methods” section “Material Selection”
95-96 — Why are the birds of different ages? How did you select them? By some parameters?
97 - On MacConkey agar — add a photo
256 - “E. coli strains isolated from fecal samples of broilers (n = 43) and turkeys (n = 17)” — What do these numbers mean?
259 - “β-lactams (7 antibiotics) and carbapenems (1 antibiotic)”
or
“Aminoglycosides (gentamicin - GEN, amikacin - AMK), Amphenicols (chloramphenicol - CHL), β-lactams (ampicillin - AMP, cefotaxime - CTX, ceftazidime - CAZ, cefoxitin - FOX, cefepime - FEP), β-lactams with clavulanic acid (cefotaxime/clavulanic acid – CTX/CA, ceftazidime/clavulanic acid – CAZ/CA), Quinolones (ciprofloxacin - CIP, nalidixic acid - NAL), Folate inhibitors (sulfamethoxazole - SMZ, trimethoprim - TMP), Carbapenems (meropenem - MRP, imipenem - IMP, ertapenem - ETP), Polimyxins (Colistin - COL), Tetracyclines (tetracycline - TET, tigecycline - TIG) and Macrolides (azithromycin — AZI)” — that's more than 8 antibiotics
339 - “livestock sector” — replace with “poultry sector”
349 — only 102 samples in 2 years? — the results will not be statistically significant
How exactly and by what principle did you take biological samples? Were the feces fresh?
Was it the same farm? Or different enterprises?
Was the bird clinically healthy or sick, or had it already been slaughtered?
What age and breed were the birds?
I strongly recommend adding a section — MATERIAL SELECTION
add housing conditions, feeding, epizootic situation, total livestock
358 — “Incubation was carried out at 44°C for 20 hours.” - Explain this temperature and cultivation time. Is the bacterium able to multiply at this temperature?
366 - “From the 24-hour primary culture obtained on agar, 3-4 colonies were transferred into 5 mL of sterile distilled water.....
You write sterile distilled water everywhere - distilled water is a priori sterile, so just “distilled water” is enough
372 - “The inoculated plates were sealed with adhesive film to prevent drying and incubated at 35°C for 20 hours.” - why is the temperature and cultivation time like this?
373 — What is the reason for such a wide range of antibiotics? Are they all used in veterinary medicine? Are they used to stimulate the growth of birds?
Correct the typos in the text
Best regards,
Reviewer
Reviewer 4 Report
Comments and Suggestions for Authors
This study investigated antimicrobial resistance in E. coli strains isolated from poultry, focusing on the presence of ESBL/AmpC producers, associated resistance genes (ampC, blaZ, blaTEM), and multidrug resistance patterns. The findings emphasize the urgent need for integrated surveillance and stewardship strategies to address rising resistance threats in the livestock sector. However, there is some confusion with the calculation of percentages throughout the manuscript, and the data presentation needs clarification. I recommend revision based on the following comments.
- Table 1 and others: Please define what the percentage values represent (e.g., percentage of total isolates?) both in the main text and in the table legends. This clarification should be consistently applied across all tables.
- Lines 119–123: I am confused by the statement, “Out of the total isolated strains, 6 (19%) were classified as MDR.” If the total number of isolates is 102, 6 does not correspond to 19%. Please clarify the basis for this percentage. Similar discrepancies appear throughout the manuscript—these should be reviewed and corrected for consistency.
- Table Legends: Expand the legends to clearly explain any color coding, highlights, or symbols used.
- Figure 1: Label the y-axis as “Number of Isolates” for clarity.
- Figure 2: The legend should explain what "1" and "0" represent. Also, mention the software or tool used to generate the matrix.
- Line 25: The statement “Gene detection revealed 76.2% ampC, 72.7% blaZ, and 58.1% blaTEM prevalence, yielding a total penetrance of 51.09%” is confusing. Please clarify the basis for these percentages—are they relative to all isolates or a subset?
- Line 19: Such “strains”. Please correct.
- Line 45: The correct phrase should be “inhibitors of bacterial cell wall synthesis,” not “inhibitors of bacterial cell walls.”
Round 2
Reviewer 1 Report
Comments and Suggestions for Authors
The authors have made significant improvements to the manuscript and have addressed many of my previous concerns. However, I still have some concerns regarding the correlation analysis, which now appears more confusing than before:
- Line 233: The authors state that cefotaxime (CTX), ceftazidime (CAZ), and cefepime (FEP) have a Pearson correlation (r) value > 0.80. However, I can also see r > 0.80 for other antibiotics such as FEP, FOX, and SMZ etc. Moreover, there appear to be discrepancies in the correlation analysis compared to the previous version, as indicated by the increased number of red-colored boxes. For example, the correlation between AMP and CTX is now shown as r = 0.99, whereas it was r = 0.61 previously. Similar discrepancies are observed between AMP and CTZ/CA, as well as CAZ/CA. Also, why have all the blue boxes suddenly disappeared in the revised version of the manuscript?
- Line 238: The authors state that antibiotics such as amikacin (AMK), colistin (COL), tigecycline (TIG), and azithromycin (AZI) exhibited very low correlations with other antibiotics. However, the current correlation plot shows strong correlations for at least three of these antibiotics with others like TET, FOX etc. Could the authors clarify or explain this apparent inconsistency?
- Figure 2: The authors mention that the results of Figure 2 are discussed in lines 270-274. However, these lines describe the clustering analysis and do not explain the correlation findings presented in the figure. Please clarify or correct the referenced text.
- Regarding a previous comment on data discrepancy on sulfamethoxazole-resistant strains in broilers-68% in the main text vs. 74% in the table: The authors responded that this was corrected in lines 114-115, but in the revised manuscript, the corrected lines appear to be 160-161. Please ensure consistency between the reported numbers and the corresponding locations in the manuscript.
Author Response
Please see the attachment and supplementary file

Reviewer 2 Report
Comments and Suggestions for Authors
The Authors have done well in reducing major corrections but there are still work to be done to improve the work. please see below my comments.
- In line 519, Authors mentioned that the Gel electrophoresis was conducted, but was not presented. Please can Authors present the gel electrophoreses in the result section or it can be moved or attached as supplementary file.
- In my previous review (comment 5), I highlighted that Authors should include company manufacturering and country of media used. It the media was not purchased, like Api 20E, MacConkey agar and MacConkey agar supplemented mentioned in the work. If the media laboratory product of the Authors, it should also be stated.
- Authors mentioned that “The lack of financial resources led to being unable to provide the positive control for each gene, but contacted the EURL-AR laboratory and we were advised to search in the literature for the number of base pairs the amplicons at the end of the PCR method must have for each gene in part”. I will suggest Authors to include a “limitation section” before the reference section in the manuscript, where some of the limitations will be mentioned. So that future researchers can benefit from the limitations the study encountered and improves the study area. The limitation was cited around “3.2. Diagnostic Odds Ratio” section which is not easy to access.
- line 559-564: Authors mentioned that “a chi-square test of 559 independence was applied to the data in Table 1. A p-value < 0.05 was considered statis- 560 tically significant. 561 For Tables 2 and 3, Fisher’s exact test was used to assess whether the observed dif- 562 ferences in antibiotic resistance and susceptibility frequencies between turkeys and broil- 563 ers were statistically significant, particularly where cell frequencies were small” the result of the chi square, and the resultant p-values should be presented in table Table 1, 2 and table 3 where statistical analysis was conducted, and result of the analysis presented in the result section and discussed.
- In discussion section, Authors should avoid overstatement without referencing/citations. See Line 377 – 379 should have a citation, 379-381 should have a citation; (variants echoes findings in 380 other researches [0, 0, 0….] that emphasize the need for careful monitoring of resistance patterns) ? “other researchers?......................mention the other researchers. Line 399 -403 should have a citation.
Authors should take time to go through the discussion section and cite/reference important statements,.
Reviewer 3 Report
Comments and Suggestions for Authors
Dear colleagues!
Thank you for your comments on my remarks - I find them comprehensive and well-founded.
Best regards,
Reviewer
Author Response
Dear Reviewer,
Thank you for your response and for taking the time to review our replies. We’re glad to hear that you found our comments clear and satisfactory.
We appreciate your input and the constructive role it played in refining our work.
Best regards,
Alex C. Moza
On behalf of the co-authors.